# PROTOTYPICAL REPRESENTATION LEARNING FOR RELATION EXTRACTION

**Ning Ding**[1][*]**, Xiaobin Wang**[2][*]**, Yao Fu**[3]**, Guangwei Xu**[2]**, Rui Wang**[2]**, Pengjun Xie**[2]**,
Ying Shen**[4]**, Fei Huang**[2]**, Hai-Tao Zheng**[1][†]**, Rui Zhang**[‡]
[1]Tsinghua University, [2]Alibaba Group, [3]The University of Edinburgh, [4]Sun Yat-sen University
`{dingn18}@mails.tsinghua.edu.cn`, `{yao.fu}@ed.ac.uk`
`{xuanjie.wxb, kunka.xgw, chengchen.xjp, f.huang}@alibaba-inc.com`
`{zheng.haitao}@sz.tsinghua.edu.cn`, `{rui.zhang}@ieee.org`

## ABSTRACT

Recognizing relations between entities is a pivotal task of relational learning. Learning relation representations from distantly-labeled datasets is difficult because of the abundant label noise and complicated expressions in human language. This paper aims to learn predictive, interpretable, and robust relation representations from distantly-labeled data that are effective in different settings, including supervised, distantly supervised, and few-shot learning. Instead of solely relying on the supervision from noisy labels, we propose to learn prototypes for each relation from contextual information to best explore the intrinsic semantics of relations. Prototypes are representations in the feature space abstracting the essential semantics of relations between entities in sentences. We learn prototypes based on objectives with clear geometric interpretation, where the prototypes are unit vectors uniformly dispersed in a unit ball, and statement embeddings are centered at the end of their corresponding prototype vectors on the surface of the ball. This approach allows us to learn meaningful, interpretable prototypes for the final classification. Results on several relation learning tasks show that our model significantly outperforms the previous state-of-the-art models. We further demonstrate the robustness of the encoder and the interpretability of prototypes with extensive experiments.

## 1 INTRODUCTION

Relation extraction aims to predict relations between entities in sentences, which is crucial for understanding the structure of human knowledge and automatically extending knowledge bases (Cohen & Hirsh, 1994; Bordes et al., 2013; Zeng et al., 2015; Schlichtkrull et al., 2018; Shen et al., 2020). Learning representations for relation extraction is challenging due to the rich forms of expressions in human language, which usually contains fine-grained, complicated correlations between marked entities. Although many works are proposed to learn representations for relations from well-structured knowledge (Bordes et al., 2013; Lin et al., 2015; Ji et al., 2015), when we extend the learning source to be unstructured distantly-labeled text (Mintz et al., 2009), this task becomes particularly challenging due to spurious correlations and label noise (Riedel et al., 2010).

This paper aims to learn predictive, interpretable, and robust relation representations from large-scale distantly labeled data. We propose a prototype learning approach, where we impose a prototype for each relation and learn the representations from the semantics of each statement, rather than solely from the noisy distant labels. Statements are defined as sentences expressing relations between two marked entities. As shown in Figure 1, a prototype is an embedding in the representation space capturing the most essential semantics of different statements for a given relation. These prototypes essentially serve as the center of data representation clusters for different relations and are surrounded by statements expressing the same relation. We learn the relation and prototype representations based on objective functions with clear geometric interpretations. Specifically, our approach assumes prototypes are unit vectors uniformly dispersed in a unit ball, and statement embeddings are centered

---

[*]Equal contribution.

[†]Corresponding author.

[‡]http://ruizhang.info

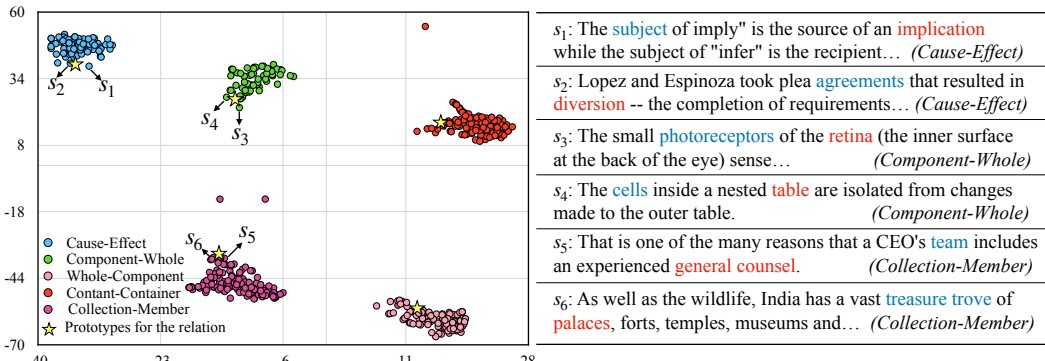

Figure 1: The $t$-SNE visualization of relation representations and corresponding prototypes learned by our model. In the right part, $s_{1:6}$ are examples of input statements, where red and blue represent the head and tail entities, and the *italics* in parenthesis represents the relation between them.

at the end of their corresponding prototype vectors on the surface of the ball. We propose statement-statement and prototype-statement objectives to ensure the *intra-class compactness* and *inter-class separability*. Unlike conventional cross-entropy loss that only uses *instance-level* supervision (which could be noisy), our objectives exploit the interactions among *all statements*, leading to a predictively powerful encoder with more interpretable and robust representations. We further explore these properties of learned representations with extensive experiments.

We apply our approach using a pretraining fine-tuning paradigm. We first pretrain a relation encoder with prototypes from a large-scale distantly labeled dataset, then fine-tune it on the target dataset with different relational learning settings. We further propose a probing dataset, FuzzyRED dataset (Section 4.3), to verify if our method can capture the underlying semantics of statements. Experiments demonstrate the predictive performance, robustness, and interpretability of our method. For predictive performance, we show that our model outperforms existing state-of-the-art methods on supervised, few-shot, and zero-shot settings (Section 4.4 and 4.2). For robustness, we show how the model generalize to zero-shot setting (Section 4.2) and how the prototypes regularize the decision boundary (Section 4.5). For interpretability, we visualize the learned embeddings and their corresponding prototypes (Section 4.4) and show the cluster clearly follow the geometric structure that the objective functions impose. The source code of the paper will be released at https://github.com/Alibaba-NLP/ProtoRE.

## 2 RELATED WORK

Relation learning could be mainly divided into three categories. The logic-based methods reason relations via symbolic logic rules, with adopting probabilistic graphical models or inductive logic systems to learn and infer logic rules orienting to relations (Cohen & Hirsh, 1994; Wang et al., 2015; Yang et al., 2017; Chen et al., 2018; Kazemi & Poole, 2018; Qu & Tang, 2019). The graph-based methods encode entities and relations into low-dimensional continues spaces to capture structure features of KBs. (Nickel et al., 2011; Bordes et al., 2013; Yang et al., 2015; Wang et al., 2014; Nickel et al., 2016; Ji et al., 2015; Lin et al., 2015; Trouillon et al., 2016; Sun et al., 2019; Balažević et al., 2019). The text-based methods have been widely explored recently, focusing on extracting semantic features from text to learn relations.

The conventional methods to learn relations from the text are mainly supervised models, like statistical supervised models (Zelenko et al., 2003; GuoDong et al., 2005; Mooney & Bunescu, 2006). As deep neural networks have gained much attention then, a series of neural supervised models have been proposed (Liu et al., 2013; Zeng et al., 2014; Xu et al., 2015; Santos et al., 2015; Zhang & Wang, 2015; Verga et al., 2016; Verga & McCallum, 2016; Li et al., 2019; Distiawan et al., 2019; Ding et al., 2019). To address the issue of the insufficiency of annotated data, distant supervision has been applied to automatically generate a dataset with heuristic rules (Mintz et al., 2009). Accompanying with auto-labeled data, massive noise will be introduced into models. Accordingly, various denoising methods have been explored to reduce noise effects for distant supervision (Zeng et al., 2015; Lin et al., 2016; Jiang et al., 2016; Han et al., 2018a; Wu et al., 2017; Qin et al., 2018a;b; Feng et al.,

2018). Further efforts pay attention to learning relations from the text in various specific scenarios, such as zero-few-shot scenarios (Levy et al., 2017; Han et al., 2018b; Gao et al., 2019; Soares et al., 2019; Ye & Ling, 2019) and open-domain scenarios (Banko et al., 2007; Fader et al., 2011; Mausam et al., 2012; Del Corro & Gemulla, 2013; Angeli et al., 2015; Stanovsky & Dagan, 2016; Mausam, 2016; Cui et al., 2018; Shinyama & Sekine, 2006; Elsahar et al., 2017; Wu et al., 2019).

However, up to now, there are few strategies of relation learning that could be adapted to any relation learning tasks. Soares et al. (2019) introduces a preliminary method that learns relation representations from unstructured corpus. And the assumption that two same entities must express the same relation is imprecise, because it does not consider the semantics of contextual information. This paper abstracts the task from the text as a metric and prototype learning problem and proposes a interpretable method for relation representation learning. Apart from the differences in methods and task scenarios, row-less universal schema (Verga & McCallum, 2016) has a similar spirit with our method, where relation type embeddings guide clustering of statements for entity pair representations and embeddings of entity pairs are regarded as an aggregation of relations.

Although the intuition that assigns a representative prototype for each class is similar to some previous studies like Prototypical Networks (Snell et al., 2017) (ProtoNet), there exist some essential distinctions between our approach and the ProtoNet. The ProtoNet computes the prototype of each class as the average of the embeddings of all the instance embeddings, that could be regarded as a non-linear version of the nearest class mean approach (Mensink et al., 2013). The idea of mean-of-class prototypes could also be traced to earlier studies in machine learning (Graf et al., 2009) and cognitive modeling and psychology (Reed, 1972; Rosch et al., 1976). In our method, prototypes and relation encoder are collaboratively and dynamically trained by three objectives, which divides the high-dimensional feature space into disjoint manifolds. The ProtoNet performs instance-level classification to update the parameters, which is not robust for noisy labels. Our method carries out a novel prototype-level classification to effectively regularize the semantic information. The prototype-level classification reduces the distortion caused by noisy labels of the decision boundary fitted by NNs. Moreover, the ProtoNet is designed for few-zero-shot learning and our methods are more like a semi-supervised pre-training approach that could be applied to supervised, few-shot, and transfer learning scenarios. Furthermore, the ProtoNet does not contain the metric learning among instances, and our method simultaneously optimizes the prototype-statement and statement-statement metrics. We also make a geometry explanation of our method (Section 3.1).

## 3 METHOD

Our method follows a pretraining-finetuning paradigm. We first pretrain an encoder with distantly labeled data, then fine-tune it to downstream learning settings. We start with the pretraining phase. Given a large-scale, distantly labeled dataset containing training pairs $(w, r)$ where $w = [w_1, ..., w_n]$ is a relation statement, i.e., a sequence of words with a pair of entities $[h, t]$ marked, $h$ being the head and $t$ being the tail, $\exists i_1, i_2, h = w_{i_1:i_2}, \exists j_1, j_2, t = w_{j_1:j_2}$, $r$ is the relation between $h$ and $t$. $r$ is a discrete label and is probably noisy. We aim to learn an encoder $\text{Enc}_\phi(\cdot)$ parameterized by $\phi$ that encodes $w$ into a representation $s \in \mathbb{R}^m$, $m$ being the dimension:

$$s = \text{Enc}_\phi(w). \tag{1}$$

A prototype $z$ for relation $r$ is an embedding in the same metric space with $s$ that abstracts the essential semantics of $r$. We use $[(z^1, r^1), ..., (z^K, r^K)], z^k \in \mathbb{R}^m$ to denote the set of prototype-relation pairs. $K$ is the number of different relations. $z^k$ is the prototype for relation $r^k$ and superscripts denote the index of relation type. Given a batch $\mathcal{B} = [(s_1, r_1), ..., (s_N, r_N)]$, $N$ is the batch size and subscripts denote batch index, the similarity metric between two statement embeddings $d(s_i, s_j)$, and between a statement embedding and a prototype $d(s, z)$ are defined as:

$$d(s_i, s_j) = 1/(1 + \exp(\frac{s_i}{||s_i||} \cdot \frac{s_j}{||s_j||})), \quad d(s, z) = 1/(1 + \exp(\frac{s}{||s||} \cdot \frac{z}{||z||})). \tag{2}$$

Geometrically, this metric is based on the *angles of the normalized embeddings restricted in a unit ball*. We will explain the geometric implication of this metric in the next section. Inspired by Soares et al. (2019), we define a contrastive objective function between statements:

$$\mathcal{L}_{\text{S2S}} = -\frac{1}{N^2} \sum_{i,j} \frac{\exp(\delta(s_i, s_j)d(s_i, s_j))}{\sum_{j'} \exp((1 - \delta(s_i, s_{j'}))d(s_i, s_{j'}))}, \tag{3}$$

where $\delta(s_i, s_j)$ denotes if $s_i$ and $s_j$ corresponds to the same relations, i.e., given $(s_i, r_i), (s_j, r_j)$, $\delta(s_i, s_j) = 1$ if $r_i = r_j$ else $0$. The computation of the numerator term forces that statements with same relations to be close, and the denominator forces that those with different relations are dispersed. When constructing the batch $\mathcal{B}$, we equally sample all relation types to make sure the summation in the denominator contains all relations. Essentially, equation 3 ensures *intra-class compactness* and *inter-class separability* in the representation space. Additionally, equation 3 contrast one positive sample to all the negative samples in the batch, which puts more weights on the negative pairs. We find it empirically more effective than previous objectives like Soares et al. (2019).

### 3.1 LEARNING PROTOTYPES

Now we discuss the objective functions for learning prototypes. Denote $\mathcal{S} = [s_1, ..., s_N]$ as the set of all embeddings in the batch $\mathcal{B}$, given a fixed prototype $z^r$ for relation $r$, we denote $\mathcal{S}^r$ the subset of all statements $s_i$ in $\mathcal{S}$ with relation $r$, and $\mathcal{S}^{-r}$ the set of the rest statements. $\mathcal{Z}^{-r}$ as the set of prototypes $z'$ for all other relations except $r$. We impose two key inductive biases between prototypes and statements: (a) for a specific relation $r$, the "distance" between $z^r$ and any statements with the same relation $r$ should be less than the "distance" between $z^r$ and any statements with relations $r' \neq r$. (b) the "distance" between $z^r$ and any statements with relation $r$ should be less than the "distance" between any prototypes $z' \in \mathcal{Z}^{-r}$ and statements with relation $r$. To realize these two properties, we define two objective functions:

$$\mathcal{L}_{\text{S2Z}} = -\frac{1}{N^2} \sum_{s_i \in \mathcal{S}^r, s_j \in \mathcal{S}^{-r}} \big[ \log d(z^r, s_i) + \log(1 - d(z^r, s_j)) \big], \quad (4)$$

$$\mathcal{L}_{\text{S2Z'}} = -\frac{1}{N^2} \sum_{s_i \in \mathcal{S}^r, z' \in \mathcal{Z}^{-r}} \big[ \log d(z^r, s_i) + \log(1 - d(z', s_i)) \big], \quad (5)$$

where equation 4 corresponds to (a) and equation 5 corresponds to (b). These objectives effectively *splits the data representations into $K$ disjoint manifolds centering at different prototypes*. We further highlight the differences between equation 4 and 5 and a conventional cross-entropy loss: there is no interactions between different statements in the cross-entropy loss that solely relies on the *instance level* supervision, which is particularly noisy under a noisy-label setting. On the other hand, our loss functions consider distances between different statements and prototypes, which exploits the interactions between statements. This type of interaction would effectively serve as a regularization to the decision boundary, as we will empirically verify in section 4.5.

Combining equations 2, 3, 4 and 5, we further give a geometric explanation of the representation space: a prototype is a unit vector (because we normalize the vector length in equation 2) starting from the origin and ending at the surface of a unit ball, and statements for that prototypes are unit vectors with approximately same directions centering at the prototype (because our objective functions 3, 4, 5 push them to be close to each other). Under the optimal condition, different prototype vectors would be uniformly dispersed with the angles between them as large as possible (because the distance metric is based on the angle in equation 2 where the minimum is at $\pi$, and we maximize distances between prototypes in equation 4 and 5). Consequently, the dataset is clustered with *each cluster centered around the end of one prototype vector* on the surface of the unit ball. The illustration of the intuition is shown in Figure 2.

Figure 2: An illustration of the geometric explanation. Stars represent prototypes and circles represent statements.

To further regularize the semantics of the prototypes, we use a prototype-level classification objective:

$$\mathcal{L}_{\text{CLS}} = \frac{1}{K} \sum_k \log p_\gamma(r^k | z^k), \quad (6)$$

where $\gamma$ denotes the parameters of an auxiliary classifier. Our prototype essentially serves as a regularization averaging the influence of label noises. We further validate this regularization in Section 4.5 by showing that it reduces the distortion of the decision boundary caused by noisy labels. Finally, with hyper-parameters $\lambda_1$, $\lambda_2$ and $\lambda_3$, the full loss is:

$$\mathcal{L} = \lambda_1 \mathcal{L}_{\text{S2S}} + \lambda_2 (\mathcal{L}_{\text{S2Z}} + \mathcal{L}_{\text{S2Z'}}) + \lambda_3 \mathcal{L}_{\text{CLS}}. \quad (7)$$

## 3.2 Fine-tuning on Different Learning Settings

We apply our learned encoder and prototypes to two typical relational learning settings: (a) supervised; (b) few-shot. Under the supervised learning setting, having the pretrained encoder at hand, we fine-tune the model on a dataset of a target domain (which could again, be distantly labeled) containing new training pairs $(w, r)$, we encode $w$ to the representation and feed it to a feed-forward classifier:

$$p(r|w) = \text{FF}(\text{Enc}_\phi(w)), \tag{8}$$

where $\text{FF}(\cdot)$ denotes a feed-forward layer, and we train it with the conventional cross-entropy loss.

Under the few-shot learning setting, we assume a slightly different input data format. Specifically, the training set consists of the relations $[r^1, ..., r^K]$ and a list of supporting statements: $s_1^k, ..., s_L^k$ for all $r^k$. Denote $\mathcal{S}^\star = \{s_l^k\}_{l=1,...,L}^{k=1,...,K}$ all supporting statements for all relations. Given a query statement $q$, the task is to classify the relation $r^*$ that $q$ expresses. During traing, we use the average distance to different prototypes as the logits of a feed-forward classifier and train the model with cross-entropy loss. During testing, we classify $q$ according to the minimum similarity metrics:

$$s_{l^*}^{k^*} = \underset{s_l^k \in \mathcal{S}^\star}{\arg\min}\, d(s_l^k, \text{Enc}_\phi(q)), \quad r^* = k^*. \tag{9}$$

Note this is equivalent to choosing the argmax of the logits because the logits are formed by the similarity scores. We note that model pre-trained by our approach performs surprisingly well even with no training data for fine-tuning, which is reported in Figure 3 and 4 in Section 4.2.

## 4 Experiments

In order to evaluate the performance of prototypical metrics learning, we conduct extensive experiments on three tasks: supervised relation learning, few-shot learning and our proposed fuzzy relation learning evaluation. We make a comprehensive evaluation and analysis of our method, as well as full-scale comparisons between our work with previous state-of-the-art methods.

### 4.1 Exprimental Details

**Pretraining dataset**   We prepare the weakly-supervised data by aligning relation tuples from the Wikidata database to Wikipedia articles. Specifically, all entities in wikipedia sentences are identified by a named entity recognition system. For an entity pair $(h, t)$ in a sentence $w$, if it also exists in the knowledge base with a relation r, then $w$ will be distantly annotated as r. After processing, we collect more than 0.86 million relation statements covering over 700 relations.

**Baseline Models**  We primarily compare our model with MTB because it is a previous SOTA model. For fair comparison, we re-implement MTB with $\text{BERT}_{\text{base}}$ (Devlin et al., 2018) and pretrain it on our collected distant data as same as our methods. We note under our setting, the reported number of MTB is smaller than its original paper because: (a) the original paper uses much larger dataset than ours (600 million vs 0.86 million); (b) they use $\text{BERT}_{\text{large}}$ encoder. In addition, multiple previous state-of-the-art approaches are picked up for comparison in all the three relation learning tasks (detailed later). For baseline models (PCNN, Meta Network, GNN, Prototypical Network, MLMAN) with publicly available source code, we run the source code manually, and if the produced results are close to those in the original paper, we would select and report the results in the original paper. For ones without open source code ($\text{BERT}_{\text{EM}}$, MTB), we re-implement the methods with $\text{BERT}_{\text{base}}$ pre-trained on our distant data and report the results. We also implement a baseline that directly trains $\text{BERT}_{\text{base}}$ on our distant data with cross-entropy loss ($\mathcal{L}_{\text{CE}}$), namely $\text{BERT}_{\text{CE}}$ in the section (the encoder is pre-trained by directly predicting the distant label for each statement).

**The IND baseline**  We additionally design a baseline, where the prototypes are pre-computed by vectorizing the extracted relation patterns independently (IND). In this way, for each statement, we pre-compute a fixed metric as its weight demonstrating the similarity with the corresponding prototype. The IND baseline is primarily for validating the effectiveness of trained prototypes over the rule-based prototypes. Given the distantly-labeled training set, we use the Snowball algorithm from Agichtein & Gravano (2000) to extract relation patterns. This algorithm takes a corpus of relational instances as inputs and outputs a list of patterns and their embeddings for each relation. After getting these patterns, we use patterns to match instances and calculate how many instances can each pattern match. For each relation label, we select the top-k patterns that match the most instances

and use the average of their embeddings as the prototype embedding. Note that these embeddings are not comparable to the sentence embeddings generated by our encoders, so the loss functions in our equation 4 and 5 are not usable for the prototypes generated here. To incorporate the information of the extracted prototype embeddings, we slightly modify equation 3 and use the similarity of instances to the embeddings to weight the instance embeddings. The modified loss function is:

$$\mathcal{L}_{\text{IND}} = -\frac{1}{N^2} \sum_{i,j} \frac{\exp(\delta(s_i, s_j)d(s'_i, s'_j))}{\sum_{j'} \exp((1 - \delta(s_i, s_{j'}))d(s'_i, s'_{j'}))}, \tag{10}$$

$$s'_i = \text{sim}(w_i, z'(w_i)) \cdot s_i, \quad s'_j = \text{sim}(w_j, z'(w_j)) \cdot s_j, \quad s'_j = \text{sim}(w'_j, z'(w'_j)) \cdot s'_j, \tag{11}$$

where $w_i$ denote the corresponding sentence to $s_i$, $z'(w_i)$ denote the correspond prototype.

**Implementation Details**     For encoding relation statements into a feature space, we adopt deep Transformers (Vaswani et al., 2017), specifically BERT, as $\text{E}_\theta$. We take the input and the output of the BERT encoder as follows: given a statement $s$, we add special entity markers to mark $M_h$ and $M_t$ before the entities, then input the statement and special markers to $\text{Enc}_\phi$ to compute final representations. Note that our framework is designed independently to the encoder choice, and other neural architectures like CNN (Zeng et al., 2014) and RNN (Zhang & Wang, 2015) can be easily adopted as the encoder. We use PyTorch (Paszke et al., 2019) framework to implement our model. All the experiments run on NVIDIA Tesla V100 GPUs. The encoder is optimized with the combination of the prototype-statement and statement-statement metrics as well as the masked language model loss with the settings as follows. For the sake of saving resources, we choose $\text{BERT}_{\text{base}}$ as the backbone instead of $\text{BERT}_{\text{large}}$, although our method sacrifices a considerable part of the performance. We select AdamW (Loshchilov & Hutter, 2018) with the learning rate of $1e - 5$ for optimization. Meanwhile, Warmup (Popel & Bojar, 2018) mechanism is used during training. The number of layers is 12, and the number of heads is 12. The hidden size is set to 768 and the batch size is set to 60. We train the model for 5 epochs in each experiment.

## 4.2   FEW-SHOT RELATION LEARNING

**Dataset** We use a large few-shot relation learning dataset FewRel (Han et al., 2018b) in this task. FewRel consists of 70000 sentences (about 25 tokens in each sentence) of 100 relations (700 sentences for each relation), on the basis of Wikipedia. We utilize the official evaluation setting which splits the 100 relations into 64, 16, 20 relations for training, validation, and testing respectively.

**Experimental settings** We use accuracy as the evaluation metric in this task. The batch size for few-shot training is 4, the training step is 20000 and the learning rate is 2e-5 with AdamW. We follow the original meaning of N-way-K-shot in this paper, please refer to (Han et al., 2018b) for details.

**Results and analysis** The results for few-shot relation learning are illustrated in Table 1. It is worth noting that we report the average score rather than the best score of each method for the sake of fairness. One observation is that the task is challenging without the effectiveness of pre-trained language models. Prototypical network (CNN) only yields 69.20 and 56.44 accuracies (%) for 5 way 1 shot and 10 way 1 shot respectively. Methods that have achieved significant performances on supervised RE (such as PCNN, GNN, and Prototypical Network) suffer grave declines. MTB learns the metrics between statements based on large pre-trained language models and yield surprising results on this task. Better performance gains by considering the semantic information by applying fixed-prototypes to MTB (IND strategy). The collaborative prototype-based methods are proven to be effective as shown in the last four rows. The results of COL ($\mathcal{L}_{\text{S2S}'} + \mathcal{L}_{\text{Z2S}} + \mathcal{L}_{\text{Z2S}'}$) indicate that the loss of MTB have little influence for the performance. Finally, our final collaborative model achieve the best performance on all the $N$-way $K$-shot settings. Our method also outperforms human in 5 way 1 shot setting (92.41 vs 92.22) and 10 way 1 shot setting (86.39 vs 85.88).

**Effect of the amount of training data** In real-world scenarios, it is costly and labor-consuming to construct a dataset like FewRel containing 70,000 positive instances. Some relations rarely appear in public articles and obtain candidate instances via distance supervision introduces much noise. Therefore, the amount of instances to annotate may much larger than 70, 000. To investigate the ability of few-shot classification when lack of high-quality task-specific training data, we intentionally control the amount of instances in training data and run evaluations with the new training set. Figure 3 and Figure 4 respectively display the accuracy of the 5-way-1-shot task given a limited number

Table 1: Few-shot classification accuracies (%) on FewRel dataset. The last block is the results of the series of our models with prototypes. We use $\mathcal{L}_Z$ to briefly indicate $\mathcal{L}_{Z2S} + \mathcal{L}_{Z2S'}$. Results with $\dagger$ are reported as published, and other methods are implemented and evaluated by us. $\uparrow$ denotes outperformance over the main baseline MTB and $\downarrow$ denotes underperformance.

| Method | 5 way 1 shot | 5 way 5 shot | 10 way 1 shot | 10 way 5 shot |
|---|---|---|---|---|
| Finetune (PCNN)$^\dagger$ (Han et al., 2018b) | 45.64 | 57.86 | 29.65 | 37.43 |
| Meta Network (CNN)$^\dagger$ (Han et al., 2018b) | 64.46 | 80.57 | 53.96 | 69.23 |
| GNN (CNN)$^\dagger$ (Han et al., 2018b) | 66.23 | 81.28 | 46.27 | 64.02 |
| Prototypical Network$^\dagger$ (Han et al., 2018b) | 69.20 | 84.79 | 56.44 | 75.55 |
| MLMAN$^\dagger$ (Ye & Ling, 2019) | 82.98 | 92.66 | 73.59 | 87.29 |
| BERT$_{EM}$ (Soares et al., 2019) | 88.70 | 95.01 | 81.93 | 90.05 |
| MTB ($\mathcal{L}_{S2S'}$) (Soares et al., 2019) | 89.09 | 95.32 | 82.17 | 91.73 |
| BERT$_{CE}$($\mathcal{L}_{CE}$) | 91.02 ($\uparrow$) | 95.40 ($\uparrow$) | 84.95 ($\uparrow$) | 91.43 ($\downarrow$) |
| IND ($\mathcal{L}_{IND}$) | 89.90 ($\uparrow$) | 95.42 ($\uparrow$) | 82.47 ($\uparrow$) | 91.55 ($\downarrow$) |
| COL ($\mathcal{L}_Z$) | 90.40 ($\uparrow$) | 94.73 ($\downarrow$) | 84.27 ($\uparrow$) | 91.58 ($\downarrow$) |
| COL ($\mathcal{L}_Z + \mathcal{L}_{CLS}$) | 91.12 ($\uparrow$) | 95.45 ($\uparrow$) | 85.10 ($\uparrow$) | 91.75 ($\uparrow$) |
| COL ($\mathcal{L}_{S2S'} + \mathcal{L}_Z + \mathcal{L}_{CLS}$) | 91.08 ($\uparrow$) | 95.52 ($\uparrow$) | 85.83 ($\uparrow$) | 92.18 ($\uparrow$) |
| COL Final ($\mathcal{L}_{S2S} + \mathcal{L}_Z + \mathcal{L}_{CLS}$) | **92.51** ($\uparrow$) | **95.88** ($\uparrow$) | **86.39** ($\uparrow$) | **92.76** ($\uparrow$) |

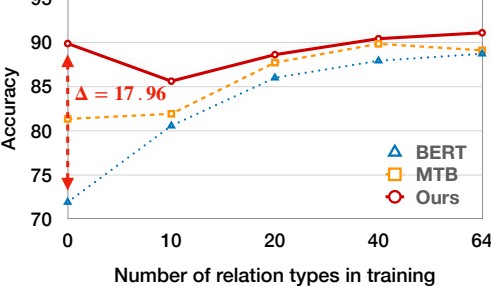

Figure 3: Impact of # of relation types

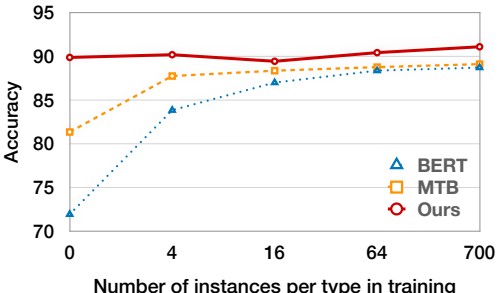

Figure 4: Impact of # of instances per type

of relation types or amount of instances of each relation in training data. As shown in the figures, pre-trained relation encoders outperform the basic BERT$_{EM}$ model when short of training data. Encoder learned with our proposed method shows large superiority over the others, when there is no training data (0), the absolute improvement of our framework is **17.96**. Besides, the accuracy increase with the increment of training data regardless of providing more relations or more instances of each relation. However, the diversity of relations is more important, since the performance in Figure 4 outperforms those in Figure 3 while the former is trained on a much smaller training data. Lack of diversity even hurts the performance when applying relation encoder which is learned with prototype-statement metric. Similar conclusions have been mentioned in Soares et al. (2019).

## 4.3 FUZZY RELATION EVALUATION

**Dataset** Aimed at evaluating relation representation learning on the noisy weakly supervised corpus, we propose a new evaluation dataset named FuzzyRED to quantitatively evaluate the capture of relational information. FuzzyRED is constructed aimed at the shortcoming of distant supervision, that is not all the statements contain the same entity pairs express the same relation. All the instances are from the weakly-supervised data and manually annotated binary labels to judge if the sentence semantically expresses the relation. For example, for "Jack was born in *London*, *UK*", even if the relation *Capital* is correct between the entities in knowledge base, the sentence does not express the *capital* relationship and will be labeled as False in FuzzyRED. The dataset contains 1000 manually annotated instances covering 20 relations. For detailed information please refer to Appendix A.

**Experimental settings** The pre-trained relation encoders are directly utilized on FuzzyRED and expected to separate the false positive instances from true positives. For each relation, we calculate the minimum of statement-statement distances among all true positives as the threshold $T$. Then for

each instance $s$ to be classified, we sample [1] $K$ (k-shot) true positive instances $s' \in \{s_1, s_2, ..., s_K\}$ and compare the average score of $a = p(l = 1|s, s')$ with $T$. If $a > T$, the instance is classified as positive. FuzzyRED is considerably challenging since it is designed for the issue of distant supervision and directly reflects the gap between semantic relations and annotation information.

**Results** The accuracies of FuzzyRED are reported in Table 2. The accuracy of labels annotated according to the DS assumption is denoted by DS, which indicates that the majority of the dataset are noise. From the experimental results, we could firstly observe that all the relation encoders suffer from the noise issue and get lower results. The impact of noise increases with the $k$ increasing. But Table 2 still shows that pre-trained relation encoders do de-

Table 2: Accuracies (%) on FuzzyRED.

| Method | 1-shot | 3-shot | 5-shot |
|---|---|---|---|
| DS | 46.3 | 46.3 | 46.3 |
| MTB | 51.3 | 50.7 | 50.7 |
| IND | 51.7 | 50.8 | 50.7 |
| COL Final | **53.2** | **52.1** | **52.1** |

tect a few noise data thus they all yield better performance than DS. Taking consideration of noise by prototypes leads to better performances, and the fixed prototypes play a small role to detect the complex semantics that express relations. Even though the COL model achieves remarkable improvement, the evaluation task is still very challenging because of the fuzziness of relation semantics.

## 4.4 SUPERVISED RELATION LEARNING

**Dataset** We use classic benchmark dataset SemEval 2010 Task 8 (Hendrickx et al., 2010) as the dataset for supervised relation learning. SemEval 2010 Task 8 contains nine different pre-defined bidirectional relations and a `None` relation, thus the task is a 19-way classification. For model selection, we randomly sample 1500 instances from the official training data as the validation set.

**Experimental settings** For evaluation, we use standard precision, recall, and F-measure for supervised RE task. Since the annotation schema of SemEval dataset differs from the data for the encoder training, we fine-tune the relation encoder in SemEval training data. The batch size is set to 20, the training epoch is 30, the learning rate is 1e-5 with Adam (Kingma & Ba, 2014). We note that the numbers for MTB are different from

Table 3: Accuracies (%) on SemEval 2010 Task 8.

| Method | P | R | F |
|---|---|---|---|
| Bi-RNN (Zhang & Wang, 2015) | - | - | 79.6 |
| Self-attention (Bilan & Roth, 2018) | - | - | 84.8 |
| BERT (Soares et al., 2019) | 86.0 | 89.1 | 87.4 |
| MTB (Soares et al., 2019) | 86.5 | 89.0 | 87.7 |
| IND | 86.5 | **89.7** | **88.0** |
| COL Final | **87.9** | 88.2 | **88.0** |

their paper because their original setting uses BERT$_{\text{large}}$ with 600 million instances and is trained on TPUs. Given the restricted resources at hand, to make fair comparison, we re-implemented their model with BERT$_{\text{base}}$ and pretrain it in our setting (0.86 million instances).

**Results** The results for supervised RE are reported in Table 3. As a classic benchmark task, results on SemEval 2010 Task 8 are already relatively high compared with other supervised RE tasks. The BERT based models outperform the traditional model significantly. By fine-tuning the relation encoder in Semeval training data, we gain 0.6 percent improvement over the original BERT model. Since the SemEval data contains rich semantic information, it is meaningful to visualize the relation representations produced by the trained relation encoders. In this part, we use $t$-SNE (Maaten & Hinton, 2008) method to project the 768-dimensional relation representations of the test set of SemEval data to

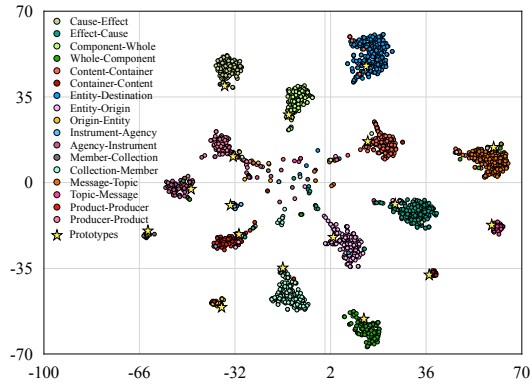

Figure 5: A $t$-SNE visualization of the relation representations and prototypes of SemEval data.

2-dimensional points. The visualization is illustrated in Figure 5, where each round marker represents a statement and a star marker represents a prototype. First, the visualization shows that statements

---

[1] For each instance, we sample 100 times and report the average accuracy.

are well clustered and the prototypes are effectively learned for all the relations. Another interesting observation is that the visualization shows visible directions and it is identical to our geometric explanation Section 3.1.

### 4.5 A TOY EXPERIMENT OF DECISION BOUNDARIES FOR PROTOTYPICAL LEARNING

In section 3.1, we mention that the prototype-level classification (equation 6) reduces the disortion of the decision boundary caused by noisy labels. In this section, we carry out a toy experiment to explore the point. We perform a simple binary classification on the iris dataset (Blake, 1998). Here, we use the instance-level linear classifier and the prototype-level classifier. For the prototype-level classifier, the decision function could be written as $g(x) = \text{sign} \|(x - z_-)\|^2 - \|(x - z_+)\|^2$, where $x$ is a data point and $z_-$ and $z_+$ represent the prototypes for two classes. This equation is also equivalent to a linear classifier $g(x) = \text{sign}(w^\top x + b)$ with $w = z_+ - z_-$ and $b = \frac{\|z_-\|^2 - \|z_+\|^2}{2}$. Prototypes $z$ are computed as the classical mean-of-class (Reed, 1972) method, thus the decision boundary could be easily depicted. We focus on the distortion of the decision boudaries, as shown in Figure 6, when the instances are wrongly labeled, the instance-level boundary is easily impacted and overfitted by noisy labels (Figure 6(b)). And the boundary of prototype-level classifier is hardly affected because of the regularization, it is an interesting phenomenon and we believe it worth further theoretical and empirical study. The results also demonstrate the potential of our model for the generalization of other standard/noisy/few-shot classification tasks.

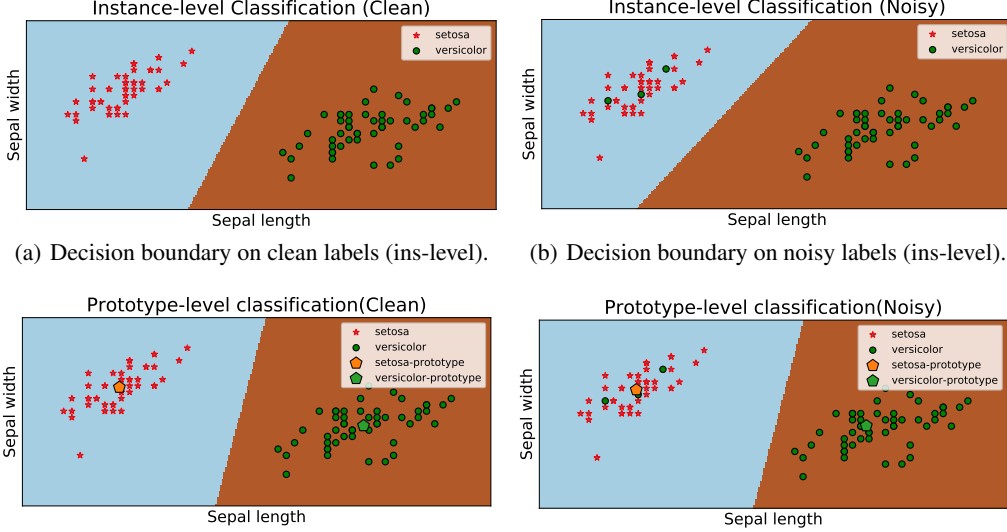

(a) Decision boundary on clean labels (ins-level).  (b) Decision boundary on noisy labels (ins-level).

(c) Decision boundary on clean labels (proto-level).  (d) Decision boundary on noisy labels (proto-level).

Figure 6: A set of toy experiments on iris dataset to illustrate the distortion of decision boundaries for instance-level and prototype-level learning.

## 5 CONCLUSION

In this paper, we re-consider the essence of relation representation learning and propose an effective method for relation learning directly from the unstructured text with the perspective of prototypical metrics. We contrastively optimize the metrics between statements and infer a prototype to abstract the core features of a relation class. With our method, the learned relation encoder could produce predictive, interpretable, and robust representations over relations. Extensive experiments are conducted to support our claim, and the method also shows the good potential of generalization.

### ACKNOWLEDGMENT

This research is supported by National Natural Science Foundation of China (Grant No. 61773229 and 6201101015), Alibaba Innovation Research (AIR) programme, the Shenzhen General Research Project (Grand No. JCYJ20190813165003837, No.JCYJ20190808182805919), and Overseas Cooperation Research Fund of Graduate School at Shenzhen, Tsinghua University (Grant No. HW2018002). Finally, we thank the valuable help of Xu Han and suggestions of all the anonymous reviewers.

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

# A   Fuzzy Relation Evaluation Dataset

## A.1   Distant Supervision

Even though our framework is adaptive for a variety of data types, weakly (distantly) supervised data is the most suitable in terms of quantity and quality. Generally, weakly supervised data is constructed on the basis of a large unstructured textual data (e.g. Wikipedia) and a large structured knowledge base (e.g. Wikidata) via the method of distant supervision. First, all the entities are identified by a named entity recognition system which could tag persons, organizations and locations, etc. Those entities form an entity set $\mathcal{E}$ that is mentioned in Section 2. Then if for a sentence $I$, the entity pair $(h, t)$ also exists in the knowledge base with a relation $r$, then $I$ will be distantly annotated as $r$. After the distant supervision, all the relations form a relation set $\mathcal{R}$.

## A.2   Motivation of FuzzyRED

The distant supervision assumption demonstrates that if two entities participate in a relation, then all the sentences that mention these two entities express that relation. While obtaining plenty of weakly (distant) supervised data, this assumption has also become the primary drawback of distant supervision. Generally, one entity pair in the distant dataset may correspond to multiple relations in different textual environments. For example, the entity pair *(Steve Jobs, California)* has three relations, which are *Place of Birth*, *Place of Work*, and *Place of Residence*. Thus, instances of these corresponding relations prone to introducing false-positive noise. We named such relations as fuzzy relations. Intuitively, conducting extraction on fuzzy relations is more challenging. To empirically evaluate if encoder trained on distant supervised could really learn the correct semantic from the noisy distant supervised corpus and perform well on Fuzzy relations, we present a novel dataset, namely, Fuzzy Relation Extraction Dataset (FuzzyRED). In this section, We will firstly introduce the construction process for FuzzyRED and report some statistics.

## A.3   Construction Process of FuzzyRED

Broadly speaking, the dataset is selected from a distant supervised corpus and judged by humans to determine whether each instance really expresses the same relation as the distant supervision labels. For instance, sentence "*Jobs* was born in *California*" really expresses the supervision label "Place of Birth" while sentence "*Jobs* lived in *California*" does not. The detailed process is as follows:

1. First, following the distant supervision assumption, we align the relation triples in Wikidata with Wikipedia articles to generate massive relation instances.

2. We count the number of corresponding relations for each entity pair, denote as $\mathrm{Count_R(pair)}$. For example, the entity pair (Steve Jobs, California) has three relations, which are *Place of Birth*, *Place of Work*, and *Place of Residence*. Larger $\mathrm{Count_R(pair)}$ implies a higher risk of introducing noise. Thus, we consider a pair whose $\mathrm{Count_R}(pair)$ larger than two as an ambiguous seed of the relations that it is involved in.

3. Then, for each relation, we count the number of ambiguous seeds, denote as $\mathrm{Count_S(rel)}$. The bigger $\mathrm{Count_S(rel)}$ is, the more likely the relation will get the wrong data, which will lead to a worse learning effect of the relation.

4. We sort the relations according to $\mathrm{Count_S(rel)}$ and select the top 20 relations as fuzzy relations of FuzzyRED. For each relation, 50 instances are randomly selected to annotate. For the specific annotation process, each annotator strictly judges whether the sentence could express the given relation according to the definition of the relation in Wikidata.

## A.4   Statistics of FuzzyRED

In FuzzyRED, there are 20 different kinds of relations, and 50 instances for each relation. As mentioned above, every instance is manually classified as true positive (TP) or false positive (FP). Table 4 shows the statistics of FP rates, which are calculated as $\frac{\mathrm{count(FP)}}{\mathrm{count(TP)+count(FP)}}$. As shown in Table 4, the average FP rate is more than 50 percent, in other words, the majority of the data are FPs. Hence, it is challenging to learn a relation extraction model from FuzzyRED if it doesn't take the

Table 4: Statistics of false positive (FP) rate of the raw data of FuzzyRED. An FP statement means that the statement does not express the relation but is distantly annotated the relation.

|  | Median | Average | Maximum | Minimum |
|---|---|---|---|---|
| False Positive Rate (%) | 49.0 | 54.1 | 94.0 | 19.0 |

noise problem into consideration. In this paper, we empirically evaluate different relation encoders by conducting a binary classification task. A better encoder is expected to distinguish TPs from FPs. Specifically, a model based on such relation encoder should predict a higher probability for TPs.

# B  ALGORITHMS

In Section 3.2, we introduce the adaptation of different downstream scenarios for our framework, which trains an encoder $\text{Enc}_\phi$. In this section, we provide the algorithms for the three relation learning scenarios.

Algorithm 1 reports the training of supervied relation learning.

---
**Algorithm 1** Training for Supervised Relation Extraction
---
**Input:** Supervised dataset $S_r = \{s_{1:n}\}$, statement encoder $\text{Enc}_\phi$
**while** not converge **do**
    Sample mini-batches $S_{batches}$ from $S_r$.
    **for** $S_{batch}$ **in** $S_{batches}$ **do**
        **for** $s$ **in** $S_{batch}$ **do**
            $\boldsymbol{s} = \text{Enc}_\phi(s)$
            $p(y|s,\theta) = \text{Softmax}(\boldsymbol{Ws} + \boldsymbol{b})$
        **end for**
        Update $\boldsymbol{W}, \boldsymbol{b}$ and $\text{Enc}_\phi$ w.r.t. $l(\theta) = \sum_{s \in S_{batch}} \log p(y|s,\theta)$
    **end for**
**end while**
---

Similarly, Algorithm 2 illustrates the training algorithm for few-shot relation learning.

---
**Algorithm 2** Training for Few-shot Relation Extraction
---
**Input:** Few-shot dataset $S_f$, statement encoder $\text{Enc}_\phi$
**repeat**
    $\mathcal{R}' = \text{SampleRelation}(N, \mathcal{R})$.
    $S = \varnothing, Q = \varnothing$
    **for** $r$ **in** $\mathcal{R}'$ **do**
        $\{s_i^r\} = \text{SampleInstance}(k, r), i \in \{1, 2...k\}$
        $q^r = \text{SampleInstanc}(1, r)$
        $S \bigcup \{s_i^r\}, Q \bigcup \{q^r\}$
    **end for**
    **for** $q_r$ **in** $Q$ **do**
        **for** $s_i^{r'}$ **in** $S$ **do**
            $\text{sim}_i^{r'} = \text{Enc}_\phi(q_r) \cdot \text{Enc}_\phi(s_i^{r'})$
        **end for**
        $\hat{r} = \text{argmax}_{r', r' \in \mathcal{R}'}(\text{sim}_i^{r'})$
        Update $\text{Enc}_\phi$ w.r.t. $\text{CrossEntropy}(r, \hat{r})$
    **end for**
**until** enough steps
---

Algorithm 3 reports the training of fuzzy relation learning. Note that FuzzyRED is designed for evaluation, so the algorithm shows the inference phase.

---

**Algorithm 3** Training for Fuzzy Relation Evaluation

---

   **Input:** FuzzyRED $S_z = \{s_{1:n}\}$, statement encoder $\text{Enc}_\phi$
   **while** not converge **do**
     **for** $s$ **in** $S_z$ **do**
       $\{s_i\} = \text{SampleInstance}(k), i \in \{1, 2, ..., k\}$
       Compute $a$ w.r.t $a = \frac{1}{k} \sum p(l = 1|s, s_i)$
       **if** $a > T$ **then**
         $s$ is true positive (TP)
       **else**
         $s$ is false positive (FP)
       **end if**
     **end for**
   **end while**

---

