# OpenReview forum: "Prototypical Representation Learning for Relation Extraction"
_ICLR.cc/2021/Conference — ICLR 2021 Poster_

### Official Review · AnonReviewer4 · 2020-10-24
**Better justification on the motivation and experiments**

**Rating:** 5
**Confidence:** 5

**Review:**

This paper presents a pre-training method for encoders (i.e., BERT) of relation extraction, leveraging distant supervision data. The main idea is to introduce a prototype embedding for each relation in the distantly generated data. The loss function to pre-train the encoders involve several terms, aiming to exploit the intra-class compactness (e.g., the contrastive loss for statement pairs) and the inter-class separability (i.e., constraining the distances between the prototype for one relation and the embeddings of the statements for the same and different relations). The paper incorporates a prototype-level classification loss. Compared to few-shot learning models (e.g., prototypical networks), the paper claims that prototype-based losses can help the model to better address the noisy label issue of distantly generated data due to the exploitation of prototype and instance interactions. The pre-trained encoder is then applied on datasets for few-shot, supervised, and fuzzy relation extraction. The proposed method achieves competitive performance with other methods. Some analysis are also conducted to demonstrate the effectiveness of the proposed model.

Overall, this paper is well written and the proposed method seems helpful for improving the performance of relation extraction on several settings according to the presented experiments. However, I have several concerns regarding the motivation and experiments of the papers as follow.


1. The paper loss terms in Equations (2), (3), and (4) seem intuitive. However, they are all based on potentially noisy assignments of labels for statements (due to distant supervision). The paper mentions that as cross-entropy loss only considers instance-level supervision, it is particularly noisy (page 4). However, it is unclear why Equations (2), (3), and (4) can better handle the noise given that the information they rely on is still inherently noisy. Relatedly, a more direct baseline for the proposed model is to just fine-tune the encoder with cross-entropy loss using the standard supervised learning setting with distantly supervised data (i.e., directly predict distant labels from statements). As this baseline seems missing, it is unclear why the proposed losses are helpful and convincing.

2. The paper uses the pre-training approach with distantly generated data. Another approach is to use multi-task learning where the encoder is trained simultaneously on both downstream task and distantly-generated data. How does this approach compare to the proposed method. Also, how does the relations in the distantly generated data differ from those in the few-shot and supervised experiments?

3. In Appendix A, the paper mentions that the encoder is also fine-tuned with masked language model loss. This seems to be an important factor (given its effectiveness in BERT); however, it is not evaluated and discussed in the main paper. How does the performance changes if this masked language model loss is not used?

4. The performance in Table 3 makes the proposed method less convincing. In particular, the proposed method (COL Final) essentially has the same performance as the baseline (IND) which raises a question about the benefits of the proposed techniques. There is no discussion between the performance difference between these two models to better justify the model. The performance gap between the proposed model and the baseline MTB seems minor too. Also, Soares et al., 2019 seem to report a different performance on SemEval than those indicated in this paper (i.e., the best score in there is 89.5). Can you clarify this detail?

---

> ### Author Response · Authors · 2020-11-18
> **Response to AnonReviewer4 (2/2)**
>
> ### Different reported numbers for MTB in Table 3
> - The primary reason for different numbers is that the implementation in their paper is much larger and requires large computational resources ($\rm BERT_{large}$ with 600 million weakly supervised data) and does not open-source their code and model.
> - While they train their model with TPU pods, the resources we have at hand are 8 Nvidia V100 GPUs. So we have to re-implement it with $\rm BERT_{base}$ on our setting with 0.86 million weakly supervised data. We have added clarification on the updated section 4.1. Some other works (e.g., Qu et al, ICML'20, arXiv:2007.02387) also re-implement MTB with $\rm BERT_{base}$, and our implementation of MTB yields comparable or higher results than theirs, which could verify the correctness of the implementation.

---

> ### Author Response · Authors · 2020-11-24
> **Response to AnonReviewer4 (1/2)**
>
> We thank the reviewer for the detailed discussions. Here is our response to the concerns of the reviewer:
> ### The effect of loss terms used in this work
> - We note that the mechanism behind the loss terms and their influence on the decision boundary through the interactions with the prototypes could be complicated, and the current interpretation is indeed intuitive.
> - Yet we still see some empirical supports in the experiments: (a) in fuzzy relation evaluation, our approach outperforms the baselines and exhibit a better capacity for recognizing the true relations from spurious surface forms; (b) Figure 6 also shows the decision boundary would be more regularized with the help from prototypes. We believe these sets of experiments would serve as the initial attempts toward a more in-depth understanding of the model.
> - Ideally, a detailed theoretical investigation of the effect of different loss terms (bounds, complexity, .etc) would give a better understanding. As such investigation would be a quite nontrivial task, we leave it to future work.
>
> ### Fine-tuning encoder with cross-entropy loss
> - Thank you for pointing this out, it is a reasonable baseline. We have added the experiments on FewRel and here are the results:
> | |Methods|5-1|5-5|10-1|10-5|
> |-|-|-|-|-|-|
>  |(a)|MTB|89.09| 95.32| 82.17|91.73|
> |(b)|Ours (change to cross entropy)| 91.02| 95.40| 84.95| 91.43|
> |(c)|Ours (in the paper)| 92.51|95.88| 86.39| 92.76|
>
> - Comparing (a) and (b), we see that the model trained on distantly supervised data with cross-entropy still outperforms the baseline MTB. Comparing (b) and (c), we validate the effectiveness of using prototypes during fine-tuning.
>
> ### To use multi-task learning
> - As pretraining and fine-tuning is currently (more or less) standard research paradigm, we follow this practice in our work.
> - Yet we do believe that with recent findings (Gururangan et al, ACL’20), multi-task learning would be effective for further boosting the performance. As this is orthogonal to our contribution, we will investigate it in the future.
>
> ### Relations of distantly generated data and the data for few-shot and supervised learning
> - Our pre-training data is constructed from Wikipedia sentences and Wikidata database (section 4.1 in the updated paper). For FewRel, it is possible that the FewRel dataset and the pretraining dataset and share some relation types because FewRel is also constructed from Wiki. For Semeval, since it is constructed from different sources, it does not share relation types with the pretraining data.
> - Moreover, as is detailed in the replies to AnonReviewer2 (bullet points discussing relation already seen during pretraining), we have made sure that the test cases do not leak in the pretraining cases, and when we further exclude the relation types in the pretraining data, the advantages of our model still hold.
>
>
>
> ### Without MLM loss
> - Thanks for pointing this out. When not using MLM loss, the results on FewRel are :
> |Methods|5-1|5-5|10-1|10-5|
> |-|-|-|-|-|
> |Ours with loss $L_Z + L_{CLS}$|91.12| 95.45| 85.10 |91.75|
> |Ours with loss $L_Z + L_{CLS} - L_{MLM}$| 90.55| 94.73| 84.47| 91.58|
>
> - The MLM indeed affects performance. Due to the time limit, other models that pretrained without MLM are still being trained now, we will add this part in the revised paper.  However, we note that adding MLM loss is (more or less) a standard practice when doing pretraining. Our contributions are indeed orthogonal to it.
>
> ### Performance in Table 3
> - The primary reason that IND and COL perform similarly on this dataset is that it actually has sufficient supervision to get good enough performance, thus not challenging enough to differentiate these models. In fact, the 0.3 difference between our model and MTB is actually a large margin on this dataset.
> - Furthermore, if we reduce the number of training instances to 20\% of the original dataset, the performance gap would be more apparent :
> |Method|Precision|Recall|F1|
> |-|-|-|-|
> |BERT| 80.19| 84.90| 82.39|
> |MTB| 80.78| 84.68| 82.60|
> |IND| 81.05| 84.91 |82.83|
> |COL|81.09| 85.83| 83.31|
>
> - Note that the two baselines are actually strong. Yet the above results consistently demonstrate the effectiveness of our approach. We design IND to demonstrate the advantages of our learned prototypes compared to the pre-computed prototypes. We have added the discussion in 4.4 in the updated paper.

---

### Official Review · AnonReviewer1 · 2020-10-25
**Nice work!**

**Rating:** 7
**Confidence:** 3

**Review:**

Summary:
The authors propose a novel method for learning prototype representation for relations which abstracts the essential semantics of relations between entities in sentences. The learned prototypes are learned based on an objective with clear geometric interpretation and have been shown to be interpretable and robust to noisy from distantly-supervised data. The method has been shown to be effective for supervised, few-shot, and distantly supervised relation extraction. The prototype embeddings are are unit vectors uniformly dispersed in a unit ball and statement embeddings are centered at the end of their corresponding prototypes. The training is done such that intra-class compactness and inter-class separability is increased. The results show that the proposed approach gives state-of-the-art results for few-shot, supervised relation extraction. The authors demonstrate the interpretability and robustness of the method.

Questions:
1. It is a bit hard to follow how increasing the number of relation types in Figure 3 improves accuracy although the problem should become harder with more relations.
2. Also, what is the intuition behind the initial decrease in performance in Figure 4 on increasing the number of instances per type although the same is not observed on other methods.

---

> ### Author Response · Authors · 2020-11-19
> **Response to AnonReviewer1**
>
> Sincerely thanks for the comments, we would answer the questions as follows:
>
> ### Increasing number of relation types means harder problem?
> - Please kindly note that in Figure 3, the increasing number of relation types means the *relation types of the training data*, and the test data will not be changed. Hence, more relation types for training means more supervision, leading to a simpler problem.
>
> ### The intuition of the decrease in Figure 4
> - The intuition is that after the pre-training procedure, our encoder has already learned intrinsic features of various relations and could produce good results on the test dataset without training data.
> - In this case, if the training data is biased (Figure 3) or insufficient (Figure 4), the model will be impacted by this lack of variety. The fluctuation of Figure 4 is smoother than Figure 3, which could also be explained by the bias of the data.
> - For other methods, we believe the initial performance is an important factor for them to not suffer the small decrease, thanks.

---

### Official Review · AnonReviewer3 · 2020-10-28
**This paper proposed a prototype based relation learning method which infers a prototype vector for each relation type to help train a better text encoder.**

**Rating:** 6
**Confidence:** 4

**Review:**

Summary:

This paper proposed a prototype-based relation learning method which infers a prototype vector for each relation type to help train a better text encoder. And authors introduced several loss functions including statement-statement similarities and statement-prototype similarities.
Experimental results over both few-shot and supervised scenarios indicate that these additional training losses improve the performance significantly. Furthermore, the author provided a dataset to indicate that the proposed method can effectively reduce the noise from distant supervision.

#####################

Reasons for score:

I vote for marginally positive. Overall, the paper is interesting and the experiments are extensive and promising. However, the main concerns listed below prevent me from giving full acceptance. And I would like to hear further responses from the author.

#####################

Main concern:

Besides promising results shown in the paper, there are some concerns about the concept of "prototype" in this paper:

1. If I understand correctly, the pretraining and fine-tuning are using the same data for all the experiments because the author didn't mention extra corpus for pretraining. Therefore prototypes are consistent (by consistent I mean there is one-to-one correspondence) with the annotated relation types in the fine-tuning (This is a little confusing because pretraining usually means using a different corpus).

2. And if prototype relation types have one-to-one mappings with relation types during fine-tuning, the key difference between the proposed method and direct training a relation classifier is that prototype vectors help to train the sentence (also called statement) encoder through S2S and S2Z losses, and then a separate classifier is trained over the statement vectors for classification. Then I'm curious why not directly add S2S and S2Z losses to the classifier? What is the reason behind generating prototype vectors?
From my understanding, "prototype" usually means that, in the meta-learning and few shot scenarios, we can generate a new model from the prototype model. However, according to formula 9, during inference, the model uses a nearest neighbor classifier without any model generation.


##############################

Some minor comments:

(1) Does formula 2 miss a negative sign for both similarity metrics? Otherwise the larger the cosine, the smaller the metric value, which is dissimilarity.
(2) it is not clear to me how to get the baseline of fixed prototypes vectors from vectors of extracted patterns. Is it average?
(3) This work shares similar spirits with Row-less Universal Schema (which has been cited in the reference) where relation type embeddings guide clustering of statement embeddings using distant supervision. A brief comparison will be useful in the related work for completeness.

---

> ### Author Response · Authors · 2020-11-19
> **Response to AnonReviewer3**
>
> Thanks for the comments, we would answer the questions as follows:
>
> ### Same data in pre-training and fine-tuning?
> - Sorry for the confusion. Please kindly note that we use different data in the pre-training and fine-tuning stages. The details of the pre-training data (which is distantly supervised with 700+ relations and 0.86 million statements) are introduced in Appendix A, we have moved it back to the main paper in section 4.1 in the latest paper.
>
> ### The prototype embeddings
> - In our paper, prototypes are embeddings that could abstract the intrinsic features of relation types, it is different from the prototype models in meta-learning, thus there is no model generation in our work.
> - The prototypes are designed to facilitate the learning of the relation encoder. With the pre-training fine-tuning paradigm, we did not add the loss terms into the classifier because the classifier only projects the relation representations into a probability space, whose inputs are dense vectors that represent relations.
> - The prototype-level classifier essentially maximizes $z_i^\intercal z_i$ and minimizes $z_i^\intercal z_j (i\neq j)$ (the detailed derivation is in the response to AnonReviewer2), which also serves for the training of the relation encoder. And the classifier does not have enough parameters to learn the massive information.
>
> ### Metrics in Formula 2
> - Thanks for reminding us, it should be dissimilarity in Formula 2, we have revised it to the ’‘distance’‘ metric in the updated paper.
>
> ### The fixed prototypes baseline (IND)
> - The purpose of using the IND baseline is to show that prototypes learned in an end-to-end, collaborative way outperform traditional rule-based prototypes.
> - Specifically, we use the Snowball algorithm (a classical relation pattern extraction algorithm) described in Agichtein and Gravano (2000) to extract relation patterns. Generally, this algorithm takes a corpus of sentences as inputs and outputs a list of relation patterns and their embeddings. The averaged embedding with the highest coverage of instances is used as the prototype.
> - We have added the details in section 4.1 in the updated paper.
>
> ### Brief comparison with Row-less Universal Schema
> - Thanks,  the spirit of the row-less universal schema has some similarities with ours. We have added a brief comparison with row-less universal schema in Section 2 in the revised paper.
> - The row-less universal schema aims to encode entity pairs as aggregate functions over the distant relation labels. Our method focuses on encoding relations of unstructured textual data by using prototypes to abstract the essential relation features. In our work, we do not pay much attention to what entities are coded into but focus on encoding the internal correlations between them.
> - The row-less universal schema see the embedding of entity pair is the aggregation of observed relations. And our method could be interestingly regarded as a ``reversed'' version, where a relation representation could embrace arbitrary entity pairs. This is quite an interesting point.
> - Besides, the specific method with prototypes and the pretraining-downstream schema is different from the row-less universal schema.

---

### Official Review · AnonReviewer2 · 2020-10-30
**Clarity and presentation issues; Unclear whether the gains are from the pre-training stage or not**

**Rating:** 4
**Confidence:** 4

**Review:**

This paper proposes a method for learning representations for relation statements as well as classes (prototypes) for relation extraction tasks. I think the key insight is that the relation statements and classes (corresponding to relation types) can be learned jointly using contrastive training objectives. The paper also proposes to use the similarity metric as 1 / (1 + exp(cos(u, v))) and claims that this will provide a clear geometric interpretation and more interpretability. In the experiments, they first train this framework on distantly-supervised data constructed from Wikidata + Wikipedia and then fine-tune it on several downstream tasks: FewRel, SemEval, and a new dataset they created focused on identifying false positives in distantly supervised data.

Overall, I think the proposed approach is quite reasonable and also seems to work well in the evaluation tasks (learning prototype embeddings instead of taking the average of instance embeddings and adopting contrastive losses between relation statements and prototypes). However, I think the paper has many clarity and presentation issues that make it difficult to evaluate the significance of the work.

First of all, I think this pre-training stage on weakly-supervised data is very crucial and the details of the data collection (which relations and how many instances have been used) should be moved to the main body of the paper instead of the Appendix. In realistic few-shot scenarios, you only have a very small number of examples for a new relation k so it is difficult to learn the r_k embedding from only a few examples. My interpretation is that for the FewRel evaluation, the relations must have been already seen in the pre-training stage (given both the weakly-supervised data and FewRel are collected based on Wikidata) unless I have misunderstood something (especially that the model can achieve a good accuracy when there is no training data used in Figure 3 & 4). For the SemEval evaluation, I assume the prototype embeddings must have been learned from the training data but it has 6k training examples so it is fine. I think this point really needs to be clarified and can be a weakness of the approach, especially in few-shot settings.

For the results in Table 1 & 3, it is not very clear to me whether the numbers of previous approaches are from their papers or re-run by the authors. This should be clarified. My main concern is the pre-training data used in this paper can be different from what has been used in (Soares et al, 2019 -- they didn’t use Wikidata and only consider Wikipedia and the links) and it makes the comparisons unfair.

The authors claimed that this similarity metric is crucial but there is no ablation study or comparison to other alternatives… How if you just compute the dot product between the two embeddings? I think the comparisons to commonly used similarity metrics need to be added to justify why this design choice is important.

I also don’t know what the L_{CLS} training loss is used for. If z_k is just a set of learnable embeddings (one embedding per relation) and if it is used to predict the relation k, isn’t it just multiplied by another set of K embeddings? What is the benefit here?

Also, I don’t understand the fixed prototype baseline (denoted as IND). What does “fixed prototypes z that are pre-computed by vectorize extracted patterns for each relation” mean?

To sum up, I can’t recommend the acceptance of the paper if the above issues cannot be addressed. I am also concerned whether the highlight of the approach is the contrastive losses and prototype embeddings, or it has to be coupled with some type of pre-training (or even specifically pre-training on distantly-supervised data).

Minor:
- Equation (7): The second S2Z should be S2Z’.

---

> ### Author Response · Authors · 2020-11-18
> **Response to AnonReviewer2 (2/2)**
>
> ### Detailed discussion on CLS loss
> - We would like to take this chance and bring up more detailed discussions. Actually, the CLS loss also has a clear geometric interpretation after a bit of math:  Let $n$ be the relation numbers and $m$ be the embedding dimension. Let $z_i$ be the embedding for the $i$-th prototype. We gather all $z_i$ into a matrix $Z \in \mathbb{R}^{m \times n}$ as the prototype embedding matrix. Suppose the classifier is a linear one with weight $W \in \mathbb{R}^{n \times m}$ (we omit the bias term $b$ for simplicity but the results still hold with $b$). So the logits of the softmax are $WZ \in \mathbb{R}^{n \times n}$. Under optimal condition, the probability matrix $P = \text{Softmax}(WZ)$ is the identity matrix $I$ (i.e., embedding $z_i$ has probability 1 to be classified to class $i$). Suppose we tie the weight $W$ with $Z$, i.e., let $W = Z^\intercal$ (a common practice in text generation literature), then the objective would effectively be maximizing $z_i^\intercal z_i$ and minimizing $z_i^\intercal z_j$ ($i \neq j$), since $z_i$ is a unit vector, so we are effectively pushing different prototype embeddings on the surface of a unit ball as far from each other as possible. Now if we do not let $W = Z^\intercal$ and optimize $W$, it is easy to show that the geometric interpretation would still hold up to a linear transformation of $W$.
> - In summary, this loss term would push the prototype embeddings as dispersed as possible on the unit ball (or equivalently up to a linear transformation), and it would be equivalent to directly minimizing the cosine similarity metrics between the unit prototype embeddings. We will further explore more distance metrics in our future research.
> - Empirically, the $\mathcal{L}_{\text{CLS}}$ loss also yields a considerable improvement of the performance, which is reported in Table 1. Here are the results on Fewrel (from the paper):
>   - |Method|5-1|5-5|10-1|10-5|
> |-|-|-|-|-|
> |COL ($L_{\text{Z}}$)|90.40|94.73|84.27|91.58|
> |COL ($L_{\text{Z}} + L_{\text{CLS}}$)|91.12|95.45|85.10|91.75|
>
>
> ### IND baseline
> - The purpose of using the IND baseline is to show that prototypes learned in an end-to-end, collaborative way outperform traditional rule-based prototypes.
> - Specifically, we use the Snowball algorithm (a classical relation pattern extraction algorithm) described in Agichtein and Gravano (2000) to extract relation patterns. Generally, this algorithm takes a corpus of sentences as inputs and outputs a list of relation patterns and their embeddings. The averaged embedding with the highest coverage of instances is used as the prototype.
> - We have added the details in the updated section 4.1.
>
> ### Highlight of our approach and core contribution
> - To be as rigorous as possible, we would only highlight the prototype itself as the core contribution of this paper because it (a) effectively enables to train predictively powerful and robust representations; (b) exhibits clear geometric interpretations of the learned representations, which would be particularly notable in the current black-box neural network era.
> - Although the prototype is coupled with the contrastive loss and pretraining in this work, instead of viewing the later two as significant contributions, we would view the contrastive loss and pretraining as a standard practice as they have already been demonstrated effective by a large amount of literature. It is also possible to design other loss types and pretraining procedures for learning prototypes, but the nontrivial contribution is still the prototype itself.
>
> ### Other similarity metrics
> - To clarify, the use of current metrics is motivated by its geometric interpretation (not from predictive performance), i.e., to restrict the embeddings on the unit ball. Metrics that do not follow this type of geometry would immediately lose the interpretability, which directly violates our original motivation and the contribution of this paper (i.e., to learn interpretable prototype representations).
> - For dot product, we have actually tested it in our preliminary experiments and find it would lead to numerical instability. To see how this would happen, we note that the dot product is unbounded. Applying it to equation 2 would lead to unbounded vector norms and the optimization would simply collapse.
> - However, other metrics on sphere geometry, e.g., the spherical distance (a distance of two points on the surface of the sphere) would be interesting alternatives. We have not yet able to test these metrics due to time limitations and will definitely test them in the future.
>
> Agichtein, Eugene, and Luis Gravano. "Snowball: Extracting relations from large plain-text collections." Proceedings of the fifth ACM conference on Digital libraries. 2000.

---

> ### Author Response · Authors · 2020-11-18
> **Response to AnonReviewer2 (1/2)**
>
> We thank the reviewer for the detailed discussions. Here is our response to the concerns of the reviewer:
>
> ### Comparisons in Table 1 and 3
> - To clarify, we emphasize that in our setting, all models (ours and MTB) are pretrained and fine-tuned on exactly the same data (our distant data). Thus the comparisons between our methods and MTB are *fair*. For models with open-source implementations, we re-run their implementations; for models without open-source implementations, we re-implement their models.
> - Specifically, for PCNN, Meta Net, GNN, ProtoNet and MLMAN, we re-run the public code. We find that the results are close to the original paper, so we report the results from the original papers.
> - For $\rm BERT_{EM}$ and MTB, we run our re-implemented version on *exactly identical data* with our model.  We note that the MTB paper uses 600 million data and $\rm BERT_{large}$ for the experiments, which would be too computationally expensive under our current resource limitation (with 8 Nvidia Tesla V100 GPUs, it is still very hard to train a $\rm BERT_{large}$ on 600 million data). Some other works (e.g., Qu et al, ICML'20, arXiv:2007.02387) also re-implement MTB with $\rm BERT_{base}$, and our implementation of MTB yields comparable or higher results than theirs, which could verify the correctness of the implementation.
>
> ### Details of data-collection
> - Our weakly-supervised data for pretraining is generated by aligning Wikipedia sentences to Wikidata database and contains 0.86 million sentences and 700+ relation.
> - We have moved the details in section 4.1 of the updated paper.
>
> ### Relations in FewRel already seen during pretraining
> we would like to take this chance to bring up more detailed discussions about what types of relation information are already seen during pretraining: (a) there might be identical test cases in FewRel accidentally included in pretraining data (which is severe and not acceptable); (b) the pretraining data may contain instances whose relation labels are included in FewRel data (but sentences not in the test data, which would be OK for real-world scenarios);
> - (a) is a severe leakage and is unacceptable. But it is possible because, as mentioned,  FewRel are collected based on Wikidata. To ensure this does not happen, we have manually excluded these cases in our experiments so there is no such leakage in our work.
> - For (b), there are some cases in the pretraining dataset sharing some relation labels with FewRel. Instead of viewing this as a weakness of our approach, we would point out that our approach still outperforms the baseline MTB by a large margin. Since the two are trained on the same data, the results still show that our approach generalizes better than the baselines.
> - We further conduct two new experiments on more strict few-shot settings to validate the generalization ability of our approach : (i) we remove training cases whole relation labels are in FewRel and fine-tune the model on FewRel; (ii) we remove training cases whole relation labels are in FewRel and further do NOT fine-tune it on FewRel.
> - For (i), our new results are (in the order 5-1, 5-5, 10-1, 10-5):
>   - MTB (pretrained with FewRel relations): 89.09, 95.42, 82.47, 91.55;
>   - COL (pretrained with FewRel relations): 92.51, 95.88, 86.39, 92.76;
>   - COL (pretrained without FewRel relations): 91.00, 95.40, 86.08, 91.87.
> - For (ii), our new results are:
>   - MTB (pretrained with FewRel relations and without finetuning): 81.32\% (5-1, in Figure 3\&4)
>   - COL (pretrained with FewRel relations and without finetuning): 90.98\% (5-1, in Figure 3\&4)
>   - COL (pretrained without FewRel relations and without finetuning): 90.00\% (5-1).
> - We observe a small performance decrease in our model, but ours trained without FewRel relations still outperforms the baseline MTB trained with FewRel relations, which further demonstrates the generalization ability of our approach on few-shot scenarios.

---

### Author Response · Authors · 2020-11-24
**General response**

Dear Reviewers,

Thanks for your reviews and suggestions. We have posted our responses to the concerns, as well as the revised paper. Could you please let us know if you have any further questions since we still could have interactions? We will respond as soon as possible.

Thanks, Authors

---

### Decision · Program_Chairs · 2021-01-07
**Final Decision**

**Decision:**

Accept (Poster)

**Comment:**

This paper proposes a method for regularizing the pre-training of an embedding function for relation extraction from text that encourages well-formed clusters among the relation types. Experiments on FewRel, SemEval 2010 Task 8, and a proposed FuzzyRed dataset show that the proposed prototype method generally outperforms prior state-of-the-art, including MTB (Soares et al., 2019), which was the strongest. The key, novel idea is to model prototype representations for target relations as part of the learning process. A contribution of the work is to show that learning prototype representations are useful in supervised deep learning architectures even beyond few-shot learning. This additional learning objective is useful as an inductive bias, and is perhaps of interest even beyond relation extraction research.

Reviewers generally found the proposed method sound and intuitive, and the original set of experiments promising. Some of the reviewers raised concerns about the setup of the experiments, including the relationship between the pre-training and target tasks, and the need for several additional baselines. The authors were able to address these concerns, and the reviewers did not raise any follow-up concerns.